# How Climate Warming Influences the Phenology of *Grapholita molesta* (Busck, 1916) (Lepidoptera: Tortricidae) in China: Insight from Long-Term Historical Data

**DOI:** 10.3390/insects15070474

**Published:** 2024-06-25

**Authors:** Haotian Bian, Wenzhuo Li, Shengjun Yu, Jianxiang Mao, Yongcong Hong, Yunzhe Song, Pumo Cai

**Affiliations:** 1College of Tea and Food Science, Wuyi University, Wuyishan 354300, China; bianhaotian@wuyiu.edu.cn (H.B.); liwenzhuo@wuyiu.edu.cn (W.L.); yushengjun@wuyiu.edu.cn (S.Y.); maojianxiang@wuyiu.edu.cn (J.M.); wyxyhyc@wuyiu.edu.cn (Y.H.); 2Key Laboratory of Biopesticide and Chemical Biology, Ministry of Education, Fujian Agriculture and Forestry University, Fuzhou 350001, China

**Keywords:** seasonal development, oriental fruit moth, overwintering, phenological response, generation

## Abstract

**Simple Summary:**

The study examined the phenological response of *Grapholita molesta* (Busck, 1916), a pest damaging Rosaceae stone fruits, to climate warming. Data on *G. molesta* population dynamics and climate in China from 1980 to 2020 were analyzed. Significant advancements were observed in the peak date of overwintering adults in northeastern, eastern, and northern China. Populations of contemporary adults in northwestern China and population peak date of overwintering adults in northern China also advanced. However, the end occurrence date of contemporary adults in northern China was delayed along with the first occurrence date of overwintering adults in northwestern China. The study highlighted spatial variations in *G. molesta*’s phenological response to climate warming across China, offering valuable insights for predicting pest infestations and enhancing pest management strategies in fruit tree cultivation.

**Abstract:**

*Grapholita molesta* (Busck, 1916), a significant pest affecting various fruits such as pears, apples, peaches, etc., is highly adaptable to changing temperatures. However, the phenological response mechanism of this pest to climate warming remains unclear. To address this issue, we collected population dynamics data of *G. molesta* in China over the years along with corresponding climate data. We analyzed five phenological indexes: the first, end, and peak occurrence dates of contemporary adults as well as the first and peak occurrence dates of overwintering adults in China. Results revealed an upward trend in the annual average temperature and average temperature of the four seasons in regions infested by *G. molesta* in eastern, northeastern, northwestern, northern, and southwestern China from 1980 to 2020. Notably, the population peak date of overwintering adults in northeastern and eastern China significantly advanced along with the first occurrence date and the population peak date of overwintering adults in northern China. Additionally, the population peak date of contemporary adults in northwestern China significantly advanced. However, the end occurrence date of contemporary adults in northern China was significantly delayed, as was the first occurrence date of overwintering adults in northwestern China. Furthermore, our study demonstrated spatial heterogeneity in the phenological response of *G. molesta* to climate warming across China. This study elucidates the phenological response of *G. molesta* to climate warming, offering valuable insights for predicting future pest infestations and informing adaptive pest management strategies in fruit tree cultivation.

## 1. Introduction

Insects—due to their sensitivity to temperature, short life cycles, strong migratory patterns, and high reproductive ability [1]—are particularly susceptible to the effects of climate change. This vulnerability leads to noticeable shifts in pest distribution and abundance even with minor changes in climatic conditions [2]. Climate change is progressively influencing the abundance of local species [3] as well as facilitating the introduction and spread of non-native species [4]. This impact is particularly evident in high-altitude and high-latitude regions and could have a greater impact on herbivores [5]. However, existing data suggest that the effects of climate change are more intricate than a simple linear response to rising average temperatures, with variations across different regions [6,7,8]. Musolin et al. [9] have emphasized that insect responses to global warming may be species- to population-specific, causing changes in distribution, phenology, abundance, population structure, and dynamics [10,11]. The magnitudes of these responses can range from undetectable to drastic and may be influenced by factors such as life history characteristics, seasons, and bioclimatic regions, sometimes yielding opposing outcomes even within the same species or population [9,12]. Climate change involves simultaneous and intricate alterations in many environmental variables, but primarily air temperature [13,14], which plays a fundamental role in regulating insect behavior, physiological functions, and evolutionary pathways [15,16]. The most recent Intergovernmental Panel on Climate Change (IPCC) report predicted that the global average temperature rise would reach or exceed 1.5 ℃ in the next 20 years [17]. The relationship between insect phenological response and climate change is complex and especially significant concerning temperature [18,19,20]. With global warming, insect phenology could undergo adaptive changes based on climate change. Therefore, it is increasingly essential to investigate how insects react to climate change, as this can significantly impact the scale and abundance of pest outbreaks and the population dynamics of agricultural pests [21,22]. This research is of great importance for both scientific understanding and practical applications.

*Grapholita molesta* (Busck, 1916) (Lepidoptera: Tortricidae), commonly known as the oriental fruit moth, is one of the major worldwide fruit-boring pests and is widely distributed around the world, except for Antarctica [23]. *Grapholita molesta* overwinters as mature fifth-instar larvae inside cocoons [24]. These overwintered larvae pupate in the spring, with adults emerging to lay eggs on host plants or near overwintering sites [25,26]. After hatching, the neonates of *G. molesta* burrow into tunnels and feed within tender twigs in the spring [27,28], while the developing larvae act as internal feeders in fruits during the mid-and late-growing seasons. This pest predominantly attacks Rosaceae stone fruits, such as apples, peaches, pears, nectarines, and cherries [29]. Cultivation regions of these plants in China—such as Anhui [30], Liaoning [31], Beijing [32], Xinjiang [33], and Tianjin [34], along with other areas including the United States [35], Brazil [36], South Korea [37]—have encountered substantial economic damage due to *G. molesta* infestations. Being a typical ectothermic animal, *G. molesta* is influenced by temperature throughout each stage of its growth and development [24,38,39]. The egg, larval, pupal, and full generation stages of *G. molesta* have specific developmental threshold temperatures of 10.39 °C, 9.95 °C, 10.97°C, and 9.80 °C, respectively [40]. Previous studies have demonstrated that developmental duration was significantly shortened with increasing temperatures ranging from 17 °C to 29 °C [41]. Fluctuations in outdoor temperature exceeding the optimal temperature of 25 °C can promote population growth and faster development [39]. Predictions indicated that climate change may result in more generations of *G. molesta*, posing economic challenges for fruit growers [35]. 

Insects are known to be sensitive to temperature, but the long-term impact of ongoing global warming on them remains uncertain due to the complex environment [18]. Therefore, it is crucial to gather empirical data on insect responses to climate change. By analyzing how representative major worldwide pests respond to climate change using empirical data, we can better understand the factors contributing to their serious damage. Unfortunately, a complete phenological database of *G. molesta* in China is lacking. However, this gap can be addressed by consulting historical records to retrieve the phenological occurrence of pests over time, as done by previous researchers [42,43]. Therefore, this study comprehensively collected the historical phenological occurrence data of *G. molesta* nationwide in China as well as the corresponding air temperature during the same period. We analyzed the interannual variation trends of *G. molesta*’s phenological data, and temperature change trends, and studied the response mechanism of *G. molesta* to climate warming. Moreover, this provided a favorable opportunity for studying the spatial heterogeneity of the impact of climate warming in different regions. This research offers practical implications for comprehending the effects of climate warming on insect pests, as well as theoretical guidance for predicting and controlling future *G. molesta*. 

## 2. Materials and Methods

### 2.1. Phenological Data Collection

Phenological data on the oriental fruit moth were gathered from historical literature sources, with a significant portion obtained from the CNKI database (http://www.cnki.net, accessed on 25 October 2023) and the Web of Science database (http://www.webofscience.com, accessed on 25 October 2023). Initially, the common and Latin names of the fruit moth species were employed as subject keywords for retrieval. Subsequently, literature from March 1982 to October 2023 that documented the occurrences, overwintering, and population dynamics of *G. molesta* in China was examined. Specific information on phenophase parameters, such as timing and geographic locations, was extracted and stored in a dedicated database. The data collection sites referenced in the literature were georeferenced using ArcGIS 10.2 (ESRI Inc., Redlands, CA, USA) and showcased on geographical maps.

### 2.2. Phenological Data Quantification and Standardization

The data collected in this study were classified based on the five most frequently recorded life cycle parameters. These parameters included the first occurrence date of overwintering adults (FOOA), the population peak date of overwintering adults (PPOA), the first occurrence date of contemporary adults (FOCA), the end occurrence date of contemporary adults (EOCA), and the population peak date of contemporary adults (PPCA). FOOA signified the date when the first overwintering adults were detected in the field, while PPOA was the date of the highest number of overwintering adults observed. FOCA represented the initial detection date of the first contemporary adult in the field, whereas EOCA indicated the final observation date of an adult in the field. Finally, PPCA was the date when the peak number of adults in the contemporary population was observed. 

Each parameter’s “day of the year” was determined by calculating the differences, in days, between the record dates of first occurrence, end occurrence, or population peak in our dataset and January 1st of that year. Some articles included vague time descriptions for these phenological parameters, such as “the beginning (middle or end) of the month,” and “the first (middle or last) ten days.” Therefore, time descriptions lacking specific dates were standardized through quantification. For instance, references to the first, middle, and last ten days of a month were quantified as the 5th, 15th, and 25th of that month, respectively. The start of a month was quantified as the first day, and the end of a month was quantified as the last day. For example, at the College of Horticulture, Qingdao Agricultural University (120.404108° E, 36.327029° N), Shandong Province, a single-plant pear orchard noted the population peak date of overwintering adults in mid-to-late April 2012. This was quantified as April 20th, with a difference of 110 days from January 1st of the same year. Therefore, the population peak date of overwintering adults of *G. molesta* in 2012 was indicated as 110 days [44].

### 2.3. Meteorological Data Collection and Analysis

The meteorological data used in this study were exclusively sourced from the China Meteorological Data Network (available at http://data.cma.cn/, accessed on 30 October 2023) dataset, which encompassed monthly and annual ground climate standard values from the China International Exchange Station. Based on the phenological information gathered for *G. molesta*, regions with a higher frequency of pest occurrences were identified and categorized into central China, eastern China, northeastern China, northwestern China, northern China, and southwestern China. The meteorological data for these regions were collected from the set of cities where phenological observations of the pest are recorded. For example, in the Central China region, cities such as Zhengzhou, Jiyuan, Luoyang, and Anyang in Henan Province and other areas have phenological records for *G. molesta*. Meteorological data from these areas were collected, and an average temperature was calculated to represent the central China region. Furthermore, the annual and seasonal average temperature of each region was calculated, and linear regression analysis was used to determine whether climate warming or cooling occurred and whether this trend of change was significant.

### 2.4. Statistics Analysis

The initial analysis involved testing the normal distribution of various datasets. The majority of datasets conformed to a normal distribution, except for two sets representing the onset and culmination of *G. molesta* contemporary adults’ occurrence. To address the non-normal distribution of these three datasets, model fitting was used to identify the most suitable model. Subsequently, a linear model was chosen as the best-fit model based on its significance. To ensure the accuracy of subsequent analyses, normalization was performed on these two datasets. Further analysis included regression analysis to explore the relationship between the year series and the change in days due to climate warming. The Pearson coefficient was employed to evaluate the correlation between seasonal mean temperature and the phenological dates. All statistical analyses were conducted using SPSS version 24.0 for Windows (SPSS Inc., Chicago, IL, USA).

## 3. Results

### 3.1. Phenological Records of G. molesta in China

As of October 2023, a total of 204 literature sources related to the phenology data of *G. molesta* were identified in the CNKI database and Web of Science (Appendix A), covering six regions: central China, eastern China, northeastern China, northwestern China, northern China, southwestern China as well as 25 provinces, including Beijing, Hebei, Shanxi, Gansu, Shaanxi, and Heilongjiang, etc. (Appendix A, Figure 1). Hebei Province has the highest number of valid records, defined by individual observation dates, succeeded in order by Shanxi, Shandong, Gansu, and Xinjiang. Due to insufficient data on the five phenological indicators in central China, this study excluded the analysis of this region.

### 3.2. Temperature Changes in G. molesta-Infested Areas in China over Time

Since 1980, a significant increase in the annual average temperature was observed across eastern China, northeastern China, northwestern China, northern China, and southwestern China. Eastern China had the highest warming rate, while the lowest was recorded by northwestern China (Figure 2a). According to research conducted between 1980 and 2020, a significant increase in the average temperature in spring, summer, and autumn occurred in five regions. The highest spring warming rate was in northern China, and the lowest was in northeastern China (Figure 2b). The highest rate of average temperature rise in summer was in the northern China region, and the lowest was in the northwestern region (Figure 2c). The region with the highest autumn warming rate was eastern China, while the region with the lowest warming rate was northern China (Figure 2d). A remarkable upward trend in the average winter temperature in five regions was shown, except for the northeastern region. The fastest warming rate was in eastern China, and the lowest warming rate was in northeastern China (Figure 2e).

### 3.3. Temporal Trend of Overwintering Adults Phenology in Different Chinese Regions

Based on the database of *G. molesta* occurrences, the average first occurrence date (FOOA) and population peak date of overwintering adults (PPOA) of *G. molesta* in different regions of China have been observed. In eastern China, the average FOOA and PPOA were recorded as 31 March and 14 April, respectively, since 1980. For northeastern China, the average FOOA and PPOA were on 1 May and 18 May, respectively. In northwestern China, the average FOOA and PPOA were 2 April and 20 April, respectively, while in northern China, these dates were 10 April and 20 April, respectively. 

The FOOA in various regions of China has shown a tendency to occur earlier than usual, particularly notable in northern China. The average yearly advances were 0.1 ± SE 0.4 days, 1.3 ± SE 0.9 days, and 0.8 ± SE 0.1 days in eastern China, northeastern China, and northern China, respectively. In contrast, the FOOA in northwestern China experienced a significant delay, with an average yearly delay of 1.0 ± SE 1.0 days. Unfortunately, data limitations prevented an analysis of the FOOA in southwestern China (Figure 3a). 

Regarding the PPOA, considerable changes have been observed over the years in various regions of China. In eastern China, northeastern China, and northern China, the PPOA advanced by an average of 0.3 ± SE 0.1 days, 1.3 ± SE 0.4 days, and 0.6 ± SE 0.1 days per year, respectively. Conversely, no significant delay trend was noted in the northwestern region, with an average delay of 0.1 ± SE 0.2 days per year. Data insufficiency prevented an analysis of the peak period of overwintering adults in southwestern China (Figure 3b).

### 3.4. Temporal Trend of Contemporary Adults Phenology in Different Chinese Regions

The study reveals that the average first occurrence date, end occurrence date, and population peak date of contemporary adults (FOCA, EOCA, and PPCA, respectively) of *G. molesta* in various regions of China have been documented. In eastern China, these dates were 13 May, 16 October, and 5 July, respectively, spanning from 1980 to 2020. For northeastern China, the average PPCA was on 2 July, while in northwestern China, the average FOCA, EOCA, and PPCA occurred on 11 May, 8 October, and 10 July, respectively. In northern China, the average FOCA, EOCA, and PPCA took place on 13 May, 23 September, and 8 July, respectively. In southwestern China, the average PPCA was observed on 14 July. 

Regarding the trends observed, the FOCA in eastern China, northwestern China, and northern China showed an insubstantial advance. On average, there was an advance of 0.4 ± SE 1.8 days, 0.9 ± SE 0.6 days, and 0.2 ± SE 0.2 days, respectively. However, inadequate data prevented the analysis of the index for the northeastern and southwestern regions (Figure 4a). 

The EOCA in northwestern China displayed a non-significant early trend with an average annual advancement of 0.6 ± SE 0.4 days. Conversely, the EOCA in eastern China and northern China exhibited a delayed trend—notably in northern China—with average annual delays of 0.9 ± SE 0.7 days and 1.0 ± SE 0.2 days, respectively. Data limitations hindered the analysis of the index for the northeastern and southwestern regions (Figure 4b). 

The study also identified an early trend in the PPCA in northeastern China, northwestern China, and northern China with significance only in northwestern China. The average annual advancements were 0.4 ± SE 0.3 days, 0.5 ± SE 0.2 days, and 0.2 ± SE 0.1 days, respectively. Conversely, PPCA in eastern China and southwestern China did not exhibit significant delays, with average delays of 0.2 ± SE 0.2 days and 0.9 ± SE 0.5 days per year, respectively (Figure 4c).

### 3.5. Phenological Response of G. molesta to Seasonal Average Temperature

In eastern China, northwestern China, and southwestern China, no significant correlations were observed between phenological indexes and mean temperatures across the four seasons (Figure 5a,c,e). However, in northeastern China, the population peak date of overwintering adults (PPOA) showed a significant negative correlation with mean temperature in autumn (r = −0.5754, Figure 5b). Conversely, the end occurrence date of contemporary adults (EOCA) in northern China exhibited a significant positive correlation with mean temperature in summer (r = 0.4172, Figure 5d). Additionally, significant negative correlations were found between PPOA and the average temperatures of spring (r = −0.3997), summer (r = −0.3313), winter (r = −0.2838), and annually (r = −0.4319, Figure 5d). Data insufficiency impeded the analysis of the first occurrence date of contemporary adults (FOCA) and EOCA in northeastern and southwestern China as well as the first occurrence date of overwintering adults (FOOA) in southwestern China, as shown in Figure 5b,e.

## 4. Discussion

With vast territory, varied topography, and various climate types, China is significantly affected by global warming [45]. Our study indicated that from 1980 to 2020, the average temperature in eastern China, northern China, northeastern China, northwestern China, and southwestern China have shown a significant warming trend in all four seasons and annually. The annual average temperature warming rates were as follows: 0.04905 ± SE 0.00482 °C/year, 0.03576 ± SE 0.00743 °C/year, 0.03324 ± SE 0.00442 °C/year, 0.04254 ± SE 0.00623 °C/year, 0.04137 ± SE 0.00389 °C/year. The ecological response of insects to climate change is visible and related to the increase in temperature [46], with their ecology and adaptability experiencing stronger and more pronounced changes, including alterations in distribution areas [47], population dynamics [48], and phenological changes [49]. Through a simple linear regression analysis of the phenological indicators of *G. molesta* and corresponding years, the results showed that in the northeastern and eastern China regions, the peak period of overwintering adult insects significantly advanced, while in northern China, the appearance period and peak period of overwintering adult insects significantly advanced, and in the northwest region, the peak period of contemporary adult insects significantly advanced. It can be seen that the results of this study were in line with the general response of insect phenology, namely, that warming temperatures will lead to an earlier insect phenology [50,51,52]. 

It is worth noting that the first occurrence of overwintering adults (FOOA) in northwestern China was significantly delayed. Generally, the effects of insects on energy demand and their metabolic rate are correlated with temperature and nutrition. As temperatures rise, insects require relatively more energy to facilitate growth and development [53,54]. This could elucidate the significant delay in the FOOA in the northwestern region. During the diapause overwintering period, *G. molesta* is affected by the increased ambient temperature, necessitating more heat to break diapause, resulting in the delayed FOOA. Additionally, rainfall also plays a role in the emergence of the overwintering generation of *G. molesta* [31], principally by regulating the temperature; hence, higher precipitation leads to lower temperatures, consequently slowing down the FOOA of *G. molesta*.

Our study found that the end occurrence date of contemporary adults (EOCA) in northern China was significantly delayed. Pearson correlation analysis indicated that a positive correlation between the EOCA in northern China and the average summer temperature. This relationship may stem from the fact that warmer summer temperatures accelerate the growth and development of *G. molesta* [39,55], shorten its development cycle, allow for more annual generations of *G. molesta*, and ultimately lead to the delayed arrival of EOCA. For example, Jung et al. (2013) indicated that the number of *G. molesta* generations per year may increase in the 2050s and 2090s due to elevated temperatures and shortened developmental periods [56]. Our findings also revealed a negative correlation between the average temperature in spring, autumn, and winter and the peak occurrence of overwintering adults (PPOA). This could be attributed to elevated spring temperatures shortening the duration of pupal growth and development [57], enabling *G. molesta* to accumulate temperature more rapidly and emerge earlier as overwintering adults. Conversely, the increased temperature during autumn and winter advances PPOA due to an accelerated metabolic rate during diapause [58,59]. This heightened metabolic rate leads to quicker consumption of insect metabolic reserves, depleting these reserves and prematurely ending diapause [60], thus advancing PPOA. Our findings showed that the average temperature in spring, autumn, and winter exhibited a significant negative correlation with PPOA. This suggested that an increase in seasonal temperature may favor the earlier emergence of overwintering adults of *G. molesta*. On the other hand, the average summer temperature showed a significant positive correlation with the EOCA. Thus, an increase in summer temperature might prolong the occurrence of *G. molesta*, potentially resulting in increased damage by the pest in the future. Therefore, exploring how the seasonal pattern of temperature changes under climate change will affect the phenological response of *G. molesta* is particularly valuable for improving our predictions of pest occurrence under global warming. 

In addition, temperature changes directly impact the host plant, potentially disrupting the delicate balance between *G. molesta* and the phenological synchronization with its host plant. Climate warming has been observed to weaken the synchrony between *G. molesta* and its respective host plants. If *G. molesta* possesses a strong adaptation ability to this weakened synchrony, it may adjust the timing of its field population’s phenological occurrences to reestablish synchrony with the host plant. This adjustment could be one reason for the shift in *G. molesta*’s phenological period associated with climate warming. Conversely, if *G. molesta* lacks a strong adaptation ability to the weakened synchrony, it might lead to the pest shifting to harm other hosts, thereby expanding the range of potential host plants for *G. molesta*. Furthermore, Souvic et al. found that the types and developmental stages of host plants significantly influenced the larval development, reproduction, and life table parameters of *G. molesta* [61]. As such, when climate change alters the phenology of its host plants, the phenology of *G. molesta* is likely to change in response.

According to the established historical database of *G. molesta*, the phenological response of *G. molesta* to climate warming showed spatial heterogeneity on a regional scale. Firstly, the phenological response of *G. molesta* varies in different regions due to temperature differences. For instance, in eastern China, the average FOOA and PPOA of *G. molesta* were observed on 31 March and 14 April, respectively. In northeastern China, the events occurred on 1 May and 18 May, respectively. In northwestern China, the average FOOA and PPOA of *G. molesta* were noted on 2 April and 20 April, respectively, while in northern China, they were recorded on 10 April and 20 April, respectively. Secondly, the long-term dynamics of *G. molesta* population showed varying trends across regions. For example, the PPCA in northeastern China, northwestern China, and northern China displayed an early trend, with only the PPCA in northwestern China showing a significant advancement, while the PPCA in eastern China and southwestern China did not indicate a significant delay. The FOOA, PPOA, and EOCA showed diverse regional changes in various parts of China. Similar phenomena have also been observed in previous studies on aphids [62] and oriental fruit flies [43]. Chen et al. discovered variations in the rapid term heat tolerance among different geographical populations of *G. molesta* and highlighted the potential for thermal acclimation to improve the rapid heat tolerance of these populations. Specifically, they observed that *G. molesta* populations in southern China exhibited greater rapid heat tolerance compared to those in central and northern China. They also identified distinct differences in heat resistance plasticity among these groups [63]. In a study by Deutsch et al. [8], it was suggested that future climate warming could lead to an increase in high-temperature occurrences, potentially reducing the average fitness of insects in tropical regions while enhancing the average fitness of insects in temperate regions. These observations underscored that the response of insects to climate change was influenced by factors such as the magnitude and nature of climate warming as well as thermal sensitivity across different geographic populations.

Temperature not only affects the life cycle of insects but also plays a crucial role in determining their geographical distribution [57]. Bale et al. [11] hypothesized that variations in growth rates and diapause requirements could significantly influence insects’ distributional responses to climate change. Species with rapid growth rates and those not entering diapause, or not requiring low temperature to initiate diapause, are likely to expand their ranges in response to warming conditions. Conversely, slow-growing species that rely on lower temperatures to induce diapause, such as certain boreal and mountainous species in the northern hemisphere, may experience range contractions. Lepidoptera remains the most intensively researched insect group in this context. For instance, butterfly species in both North America and Europe have been observed to shift their ranges northward and to high elevations as a result of rising temperatures [64,65,66,67,68]. Similar to other lepidopteran insects, the dispersion of *G. molesta* is highly influenced by global warming [69,70]. Therefore, it is imperative to implement comprehensive pest control measures that consider the phenological changes of *G. molesta*. Understanding the long-term effects of climate warming on insects is an urgent issue that holds significant implications for both scientific research and practical application. As the climate changes, not only might pest problems evolve, but the effectiveness of prevention and control measures may also change. For instance, in South Korea, the population dynamics of *G. molesta* are similar to those in China [37], likely due to the comparable climate conditions between the two countries. Therefore, orchard owners in regions where the emergence of overwintering adult *G. molesta* occurs significantly earlier or later than usual must closely monitor the insects’ phenological trends and implement timely prevention and control measures. Such a proactive approach can effectively minimize the damage caused by *G. molesta* during their peak activity period.

In this study, the range of variability was reduced by utilizing average temperature data, albeit overlooking the potential impact of extreme temperatures on *G. molesta*. However, several studies have highlighted the significant influence of extreme temperatures on *G. molesta* [71,72]. Therefore, we advocate for future research focusing on understanding the response mechanisms of *G. molesta*’s phenology to extreme temperatures. It is important to acknowledge that the phenological index data collected in this study have limitations. In China, the lack of access to long-term monitoring data on *G. molesta* occurrences has necessitated reliance on information from literature databases. Consequently, the population dynamics data used in this study is not directly derived from field surveys but from historical literature, introducing the possibility of gaps and discontinuities in the annual data relationships. Additionally, inconsistencies in time information and measurements as well as variations in field phenological investigation methods have resulted in systematic errors and compromised data quality. Therefore, meticulous attention to data collection, standardization, and analysis is vital to mitigate uncertainties associated with historical literature data. 

## 5. Conclusions

Our research has identified a significant upward trend in the average and seasonal temperatures across various regions of China impacted by *G. molesta* from 1980 to 2020. Notably, we observed an advancement in the peak dates of overwintering adult populations in northeastern and eastern China, as well as the first occurrence and peak population dates in northern China. Additionally, the peak date of contemporary adults in northwestern China advanced, while the end occurrence dates in northern China were delayed. Furthermore, the first occurrence date of overwintering adults in northwestern China experienced a significant delay. Our findings also revealed spatial heterogeneity in the phenological response of *G. molesta* to climate warming across China. These findings hold substantial significance for understanding the complex dynamics of pest species in a changing climate. They underscore the need for adaptive management strategies tailored to local climatic conditions, which can inform pest control measures that are both effective and environmentally sustainable. Furthermore, long-term monitoring studies could provide valuable data for validating and improving predictive models, enabling more accurate forecasting of *G. molesta* outbreaks and their economic impacts. 

## Figures and Tables

**Figure 1 insects-15-00474-f001:**
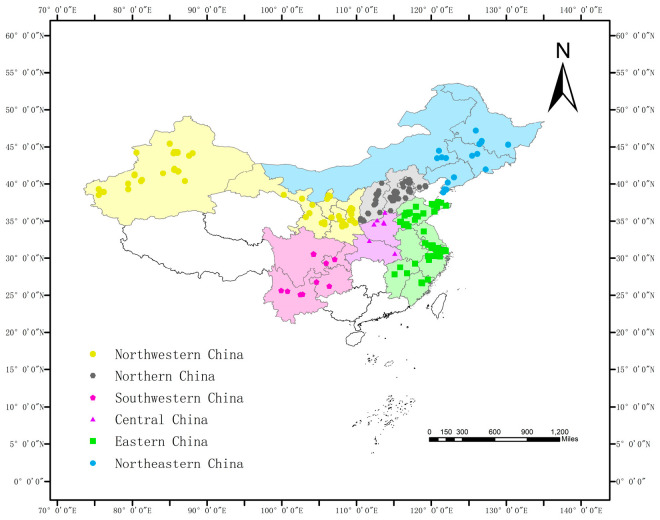
The collection sites of phenological records of *G. molesta* in different regions in China.

**Figure 2 insects-15-00474-f002:**
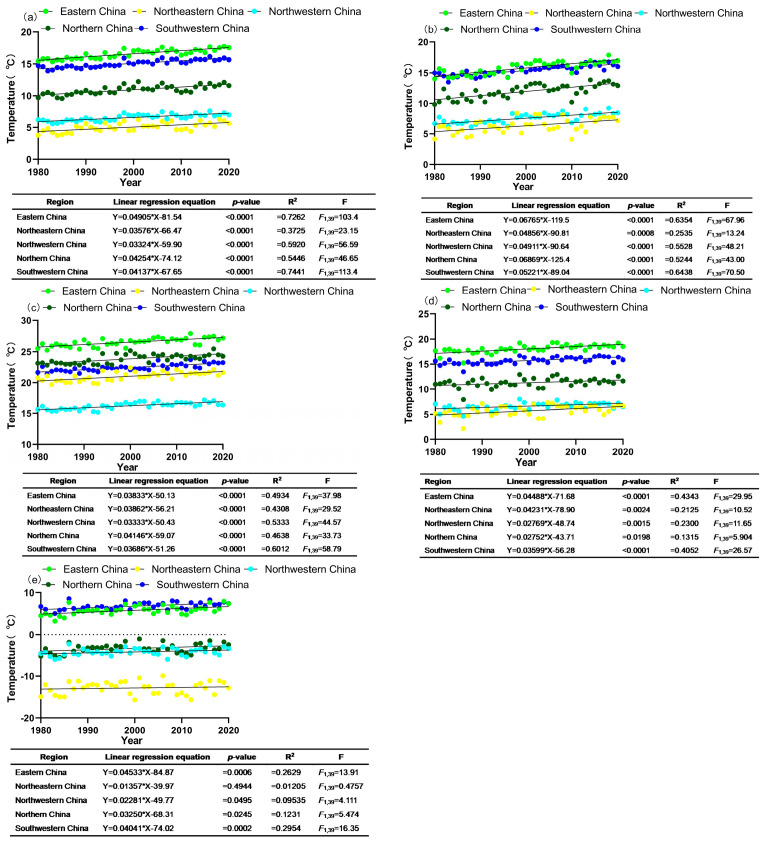
The annual average temperature and seasonal average temperature rise in five different regions of China, namely eastern, northeastern, northwestern, northern, and southwestern China over the past 40 years. (**a**) Annual mean temperature, (**b**) spring mean temperature, (**c**) summer mean temperature, (**d**) autumn mean temperature, (**e**) winter mean temperature. The data are represented by small circles that indicate the temperature record of a specific year, while the solid line shows the trend in temperature change.

**Figure 3 insects-15-00474-f003:**
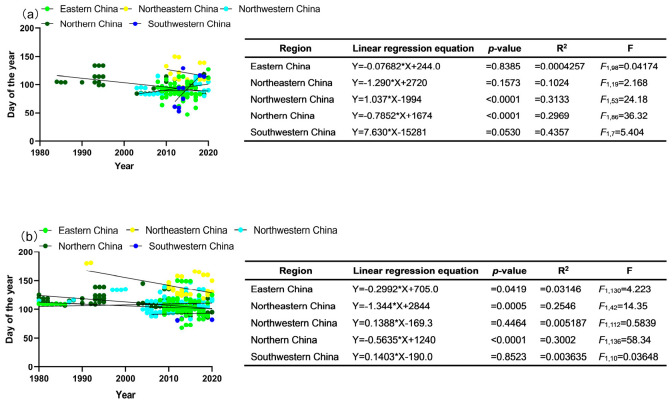
Linear regressions between the first occurrence date (**a**) and population peak date (**b**) of overwintering adults, and time (years) for *G. molesta* in five regions in China. The solid lines represent the changing trends of the phenophase parameters, while small circles indicate various phenological records.

**Figure 4 insects-15-00474-f004:**
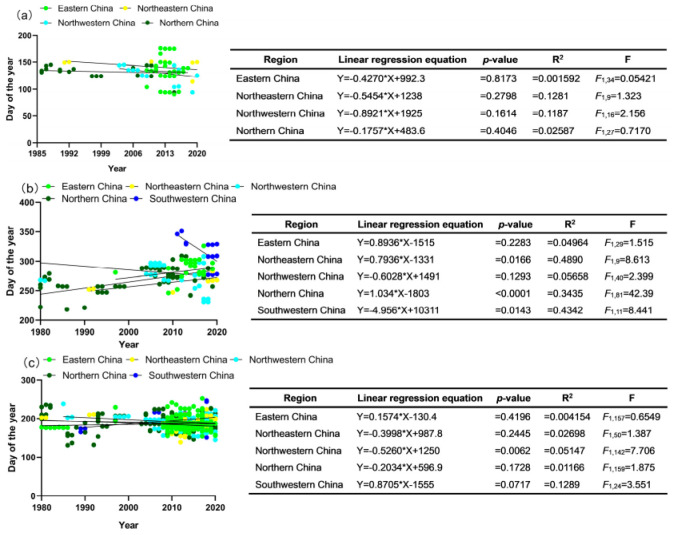
Linear regressions between the first occurrence date (**a**), end occurrence date (**b**), and population peak date (**c**) of contemporary adults and time (years) for *G. molesta* in five regions in China. The solid lines represent the changing trends of the phenophase parameters, while small circles indicate various phenological records.

**Figure 5 insects-15-00474-f005:**
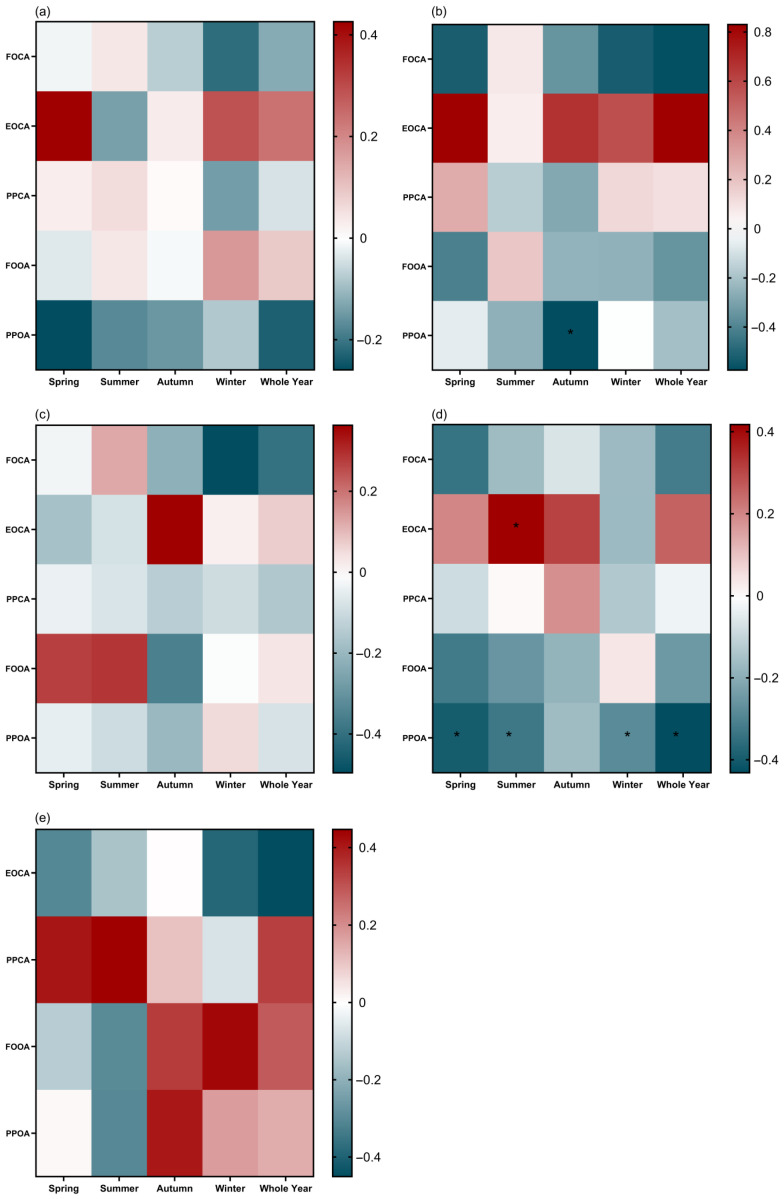
The correlations between the phenological parameters of *Grapholita molesta* and temperatures in (**a**) eastern China, (**b**) northeastern China, (**c**) northwestern China, (**d**) northern China, and (**e**) southwestern China. The black asterisk indicated the Pearson correlations were significant at the levels of *p* ≤ 0.001.

## Data Availability

The raw data supporting the conclusions of this article will be made available by the authors on request.

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
