# Peer review of "How Climate Warming Influences the Phenology of Grapholita molesta (Busck, 1916) (Lepidoptera: Tortricidae) in China: Insight from Long-Term Historical Data"

_insects, 2024, doi:10.3390/insects15070474_

Round 1
Reviewer 1 Report
Comments and Suggestions for Authors
This study elucidates the phenological response of G. molesta to climate warming, with a view to predicting future pest infestations and informing adaptive pest management strategies in fruit tree cultivation. Well written comparing population parameters to temperature variable to produce information likely to be of use in predicting occurrence. I have no criticism of the data or analysis presented-however more information about the pest would be useful. More information about the pest on world stage in the introduction and then more discussion of the implications across a broader geography than limiting to China. It is native to China but has apparently spread to most other continents-this should be stated and hence this analysis may be relevant to other parts of its distribution. Are changes in distribution identified in China from the data presented here? Also the impact of pests is related to the phenology of host plants-if available comment could be made regarding implications of relative changes in host plant phenology eg availability ‘of tender sprigs’.
Analysis reveals responses of various phenological parameters but can this be related to an overall likely impact on occurrence?
L 332 Is current information available on different numbers of annual generations of G. molesta observed throughout its wide distribution?
The discussion includes a lot of restatement of results for the various attributes (FOOA, PPOA etc). A more general interpretation of the meaning or implication of these for orchardists would be useful.
L 379 spread influenced by global warming-please expand
L 380 ff “population structure of G. molesta aligns with ecological strategies and evolutionary models, which facilitate its successful regional expansion” please expand to explain without reader needing to access ref 60.
Author Response
Response to reviewers 1
Dear reviewers:
Thanks very much for taking your time to review this manuscript. I really appreciate all your comments and suggestions! Please find my itemized responses in below and my revisions/corrections in the re-submitted files.
Question: More information about the pest on world stage in the introduction and then more discussion of the implications across a broader geography than limiting to China.
Answer: Thank you for your feedback. We have supplemented the corresponding sections accordingly.
Question: It is native to China but has apparently spread to most other continents-this should be stated and hence this analysis may be relevant to other parts of its distribution. Are changes in distribution identified in China from the data presented here?
Answer: Thank you for your suggestion. The literature we have collected is solely about papers with phenological records of this pest, not papers describing the distribution of the pest. I think it should be difficult to identify changes in the distribution range of this pest in China.
Question: Also the impact of pests is related to the phenology of host plants-if available comment could be made regarding implications of relative changes in host plant phenology eg availability ‘of tender sprigs’.
Answer: Thank you very much for your constructive feedback. We have added relevant content in the corresponding chapters.
Question: Analysis reveals responses of various phenological parameters but can this be related to an overall likely impact on occurrence?
Answer: Your question is a good one. In our study, the phenological parameters measured are the most commonly recorded field occurrence data of the G. molesta in over 200 historical documents, and we believe they should reflect the occurrence of this pest in orchards. For example, the first appearance period refers to the date when the adult G. molesta is first monitored in the orchard.
Question: The discussion includes a lot of restatement of results for the various attributes (FOOA, PPOA etc). A more general interpretation of the meaning or implication of these for orchardists would be useful.
Answer: The feedback you provided us is quite valuable, in the discussion, we summarized the results of these indicators briefly and explained them one by one without reiterating the results too much. And we have added relevant sections.
Lines 332: Is current information available on different numbers of annual generations of G. molesta observed throughout its wide distribution?
Answer: Due to the significant fluctuations in the population of G. molesta in the short term (within a few years), we have not yet found any reports on this aspect. However, we have added evidence of an increase in the generation of G. molesta under future climate warming conditions.
Lines 379 spread influenced by global warming-please expand
Answer: We appreciate your suggestions and have rewritten this section.
Lines-380: population structure of G. molesta aligns with ecological strategies and evolutionary models, which facilitate its successful regional expansion” please expand to explain without reader needing to access ref 60.
Answer: We deleted this sentence because it doesn't match the content above.
Reviewer 2 Report
Comments and Suggestions for Authors
General comments
The research tries to determine the relationship of changes in climatological data, mainly temperature, and key phenological events for the dynamics of G. molesta. SOme minor corrections are needed to publish the papers, suggestions are to add the limitation of inferences due to the variation in the range of the predictor variables. Also, analysis at the whole country level is not recommended unless a previous analysis of regions show no differences in the slopes.
Materials and methods
Describe in more detail what is a phenological record
Describe how meteorological was collected and analyzed
Explain how close/distant the meteorological data was from the pest observation records.
Results and discussion
Need to consider if results for the whole China have some meaning. The authors did not test for equality of slopes across regions before making a lumped analysis. Therefore, to present results for the whole China, a previous analysis of no differences between regions should be performed.
Tables
Reorder table 1. There is redundant information
Figures
Plots for the whole of China does not make sense because of differences in climatic parameters across the regions.

Check for the use of short instead of advancement in summary
Author Response
Response to reviewers 2
Dear reviewers:
Thank you for your letter and for the reviewers’ comments concerning our manuscript. Those comments are all valuable and very helpful for revising and improving our paper, as well as the important guiding significance to our researches. The main correction in the paper and the responds to reviewer’s comments are as following:
Question: Describe how meteorological was collected and analyzed
Answer: Thank you for your suggestion. We have added relevant content to the materials and methods section.
Question: Explain how close/distant the meteorological data was from the pest observation records.
Answer: This section is difficult to add. Since there are too many phenological observation points in each province, it is hard to describe their locations relative to the weather stations one by one.
Question: Check for the use of short instead of advancement in summary
Answer: Thank you for your suggestion. The use of the word "advancement " here is appropriate; we are unable to calculate whether this period has been shortened or extended.
Line 11-13: You mean the date to peak has been shortened?
Answer: Thank you very much for your question. To be precise, it should be that the date required to reach the peak period is shorter.
Lines 75: Not all of these fruits are Rosaceae, like papaya
Answer: We apologize for our negligence and have deleted it from the article.
Lines 162: Explain what is a phenological record in materials and methods. It is a set of observation dates or a single observation date?
Answer: Thank you for your suggestions. It is a single observation date. However, we are uncertain about the most appropriate place to include it within the materials and methods section. We think it is appropriate mentioned in first paragraph in Results section.
Lines 162: 4th column is redundant. Total should be in the last row
Answer: Thank you for your advice. We think this is an excellent suggestion. We have revised according to your suggestions. Furthermore, we have removed this table to Supplementary Materials.
Line 167: Do not repeat information in text and tables. Make reference to values in table
Answer: Thank you for your suggestion. We have rewritten the content of this section.
Line 231: What do you mean for Change of days, they are not the days of the year?
Answer: It is mentioned that Each parameter's "change of days" was determined by calculating the differences, in days, between the record dates of first occurrence, end occurrence, or population peak in our dataset and January 1st of that year in our materials and methods.
Line 297: The color scale is confuse, because it starts from the lowest r value which is negative. You should not use a sequential palette, use a diverging palette, centered in zero.
Answer: Thank you for your suggestion. We have redrawn the relevant charts with 0 as the baseline.
Line 303-305: what is the / a?
Answer: “a” means to “year”. We have revised it to “year”.
Line 353: Thus, there is no point in present results for the whole China.
Answer: Thank you for your valuable feedback. We have removed relevant content concerning the whole China from the entire text.
Line 398: The quality of the trends strongly depend of the range of observed points. Some time series appear to have a range between 2000 and 2020 but for others, the range is half or less than the study range. Therefore, the interpretation of the slopes should be taken with care.
Answer: Thank you very much for your suggestion. This is also a limitation of this study. In some years, there is a lack of historical literature on the occurrence of G. molesta. we have a description of the limitations of missing year records in line 393.
Reviewer 3 Report
Comments and Suggestions for Authors
The research topic of the authors is relevant. The scope of work is large. However, there are some comments on the presented material.
Title: I suggest:
How Climate Warming Influences the Phenology of Grapholita molesta (Busck, 1916) (Lepidoptera: Tortricidae) in China: Insight from Long-Term Historical Data
Keywords: climate warming; oriental fruit moth; overwintering; phenological response; China
Keywords must not repeat the words from the title because they are necessary for search systems.
I suggest removing „climate warming” and “China” from keywords and adding “seasonal development”, “generation” ....
The full name of the insect must be mentioned in the title, abstract, introduction – Grapholita molesta (Busck, 1916)
line 88: ecological environment? May the environment be non-ecological?
line 96: climate temperature? – air temperature?
Figure 2. Did you determine the season limits by calendar dates? Spring –March- May, Summer – June-August, and so on? It seems to me the season limits must be determined by astronomic dates (spring equinox - summer solstice - autumn equinox - winter solstice) or according to phenological criteria (spring begins after the stable transition of temperature over 0°C, summer – after the stable transition of temperature over 15°C, autumn – the stable transition of temperature below 15°C, and winter – the stable transition of temperature below 0°C. The dates of such phenological events shift with warming, and both plants and insects synchronize their development with these events and not with calendar dates.
Questions:
– Were there the same number of generations of G. molesta in different regions of China and different years?
– Did the generations overlap?
– How might this affect the dates of peak adult numbers?
It seems to me that calculating correlation coefficients between temperature and the dates of phenological events is a mechanistic approach.
The larvae should hatch when food is available. Therefore, evolutionarily, their cycles have adapted to this, and the relationship between the rate of development of caterpillars and temperature must be considered for the period after their hatching. It is known that the rate of insect development at the end of the spring and the first half of summer increases with temperature. In the second half of summer, due to the photoperiodic reaction, the development of insects may even slow down, since the insect must meet the winter in the most sustainable stage. If a larva accelerates development in the fall, pupates, and develops into an adult, the adult will not survive.
These comments do not detract from the value of the work but do provide an opportunity for the authors to think about making deeper use of the available data.
Comments on the Quality of English Languageminor editing
Author Response
Response to reviewers 3
Dear reviewers:
Thank you as the reviewer for commenting on our manuscript. We are confident that these comments will be of great help in improving the quality of our manuscript. The main correction in the paper and the responds to reviewer’s comments are as following:
Question: Title: I suggest: Keywords must not repeat the words from the title because they are necessary for search systems.
I suggest removing „climate warming” and “China” from keywords and adding “seasonal development”, “generation” ....
The full name of the insect must be mentioned in the title, abstract, introduction – Grapholita molesta (Busck, 1916)
Answer: We believe that your feedback is very meaningful, and we have made modifications to it.
Line 88: ecological environment? May the environment be non-ecological?
Answer: Thank you for your suggestions. We have deleted the “ecological” before “environment”
Line 96: climate temperature? – air temperature?
Answer: Thank you for your question. We believe that using air temperature in the description here is more accurate.
Question: Did you determine the season limits by calendar dates? Spring –March- May, Summer – June-August, and so on? It seems to me the season limits must be determined by astronomic dates (spring equinox - summer solstice - autumn equinox - winter solstice) or according to phenological criteria (spring begins after the stable transition of temperature over 0°C, summer – after the stable transition of temperature over 15°C, autumn – the stable transition of temperature below 15°C, and winter – the stable transition of temperature below 0°C. The dates of such phenological events shift with warming, and both plants and insects synchronize their development with these events and not with calendar dates.
Answer: Yes, thank you for your constructive suggestions. We divide spring, summer, autumn, and winter according to March May as spring, June August as summer, and so on. Firstly, we believe that what you said is quite accurate, but implementing this approach is almost impossible. The astronomical dates vary each year, making it very difficult to collect such climate data. Due to our survey interval being 40 years and the possibility of significant temperature fluctuations between years, temperatures reaching 0 ℃ or above in early March may occur in early April in the second year. Therefore, we first explored the trend of temperature changes over time, and then explored the trend of phenological changes over time. In addition, we also explored the impact of seasonal average temperature on the corresponding phenological indicators through Pearson.
Question: Were there the same number of generations of G. molesta in different regions of China and different years?
Answer: The peach fruit moth undergoes multiple generations within a year, with the specific number of generations depending on local temperature conditions. For example, in Northeast China, North China, and the Northwest region, it can have 3-4 generations per year; while in Shandong, Henan, and the southern part of Shaanxi, it may have 4-5 generations; in southern regions, the number can even reach 6-7 generations. The occurrence of generations varies in different areas. In areas where only peach trees are planted, the generations of the peach fruit moth occur more uniformly, with each stage appearing sequentially, making it easy to observe outbreak periods. However, in areas where peaches and pears are interplanted, there is a noticeable overlap of generations, meaning eggs, larvae, and adults can appear simultaneously.
Question: Did the generations overlap?
Answer: Yes, according to the historical literature we searched, almost all regions where G. molesta occur will experience generational overlap. See above response.
Question: How might this affect the dates of peak adult numbers?
Answer: First of all, under the environmental conditions of high temperature stress, the egg survival rate of G. molesta will be reduced, so this may reduce the number of peak periods of G. molesta to a certain extent. Similarly, when the temperature rises in the suitable range of its growth and development, the growth and development of G. molesta will be accelerated, and G. molesta will be able to breed more offspring in the same time, and the number of peak periods will be expanded. For details, see the discussion.
Question: It seems to me that calculating correlation coefficients between temperature and the dates of phenological events is a mechanistic approach.
Answer: We believe that your viewpoint is also reasonable, but we have also found that previous studies have used similar methods to study the phenological response of aphids or other insects.
Refer to: Wu Y, Li J, Liu H, et al. Investigating the impact of climate warming on phenology of aphid pests in China using long-term historical data[J]. Insects, 2020, 11(3): 167.
Gordo O, Sanz JJ (2005) Phenology and climate change: A long-term study in a Mediterranean locality. Oecologia 146(3):484–495.
Dingemanse NJ, Kalkman VJ (2008) Changing temperature regimes have advanced the phenology of Odonata in the Netherlands. Ecol Entomol 33(3):394–402.
Question: The larvae should hatch when food is available. Therefore, evolutionarily, their cycles have adapted to this, and the relationship between the rate of development of caterpillars and temperature must be considered for the period after their hatching. It is known that the rate of insect development at the end of the spring and the first half of summer increases with temperature. In the second half of summer, due to the photoperiodic reaction, the development of insects may even slow down, since the insect must meet the winter in the most sustainable stage. If a larva accelerates development in the fall, pupates, and develops into an adult, the adult will not survive.
Answer: Thank you for your constructive suggestions. the pear fruit borer belongs to the short day diapause type. In some high latitude areas, the photoperiod may have reached the critical photoperiod of the pear fruit borer by the end of summer. Most insects have already entered diapause and overwintering. So, in some low latitude areas with sufficient food, is it possible that there is also accelerated development and adult development in early autumn? Before, we have also considered the impact of photoperiod on the peach fruit moth, but it is difficult to collect data on this factor, and therefore challenging to evaluate.